# An auditory cortical-striatal circuit supports sound-triggered timing to predict future events

**Harini Suri[1], Karla Salgado-Puga[1], Yixuan Wang[1], Nayomie Allen[1], Kaitlynn Lane[1], Kyra Granroth[1], Alberto Olivei[1], Nathanial Nass[1], Gideon Rothschild**[1,2]*

**1** Department of Psychology, University of Michigan, Ann Arbor, Michigan, United States of America, **2** Kresge Hearing Research Institute and Department of Otolaryngology – Head and Neck Surgery, University of Michigan, Ann Arbor, Michigan, United States of America

* gid@umich.edu

## Abstract

A crucial aspect of auditory perception is the ability to use sound cues to predict future events and to time actions accordingly. For example, the sound of an approaching vehicle signals when it is safe to cross the street; distinct smartphone notification sounds reflect a call that needs to be answered within a few seconds, or a text that can be read later. Other animals similarly use sounds to plan, time and execute behaviors such as hunting, evading predation and tending to offspring. However, the neural mechanisms that underlie sound-guided prediction of upcoming salient event timing are not well understood. To address this gap, we employed an appetitive sound-triggered reward time prediction behavior in head-fixed mice. We find that mice trained on this task reliably estimate the time from a sound cue to upcoming reward on the scale of a few seconds, as demonstrated by learning-dependent well-timed increases in predictive licking for reward. Moreover, mice showed a dramatic impairment in their ability to use sound to predict delayed reward when the auditory cortex was inactivated, demonstrating its causal involvement. To identify the neurophysiological signatures of auditory cortical reward-timing prediction, we recorded local field potentials during learning and performance of this behavior and found that the magnitude of auditory cortical responses to the sound prospectively encoded the duration of the anticipated sound-reward time interval. Next, we explored how and where these sound-triggered time interval prediction signals propagate from the auditory cortex to time and initiate consequent action. We targeted the monosynaptic projections from the auditory cortex to the posterior striatum and found that chemogenetic inactivation of these projections impaired animals' ability to predict sound-triggered delayed reward. Simultaneous neural recordings in the auditory cortex and posterior striatum during task performance revealed coordination of neural activity across these regions during the sound cue predicting the time interval to reward. Collectively, our findings identify an auditory cortical-striatal circuit supporting sound-triggered timing-prediction behaviors.

**Data availability statement:** The data and analysis code are freely available without restriction on figshare, doi: 10.6084/m9.figshare.29033654.

**Funding:** This work was supported by a National Institute of Neurological Disorders and Stroke grant R01NS129874 (G.R.) (www.ninds.nih.gov) and an Alzheimer's Association Research Grant 21-850571 (G.R.) (www.alz.org). Funders had no role in the study design, data collection and analysis, decision to publish, or preparation of the manuscript.

**Competing interests:** The authors have declared that no competing interests exist.

**Abbreviations:** AC, auditory cortex; CNO, clozapine-N-oxide; LFP, local field potential; MGB, medial geniculate body; PBS, phosphate-buffered saline; PFC, prefrontal cortex; PLR, predictive licking ratio; pStr, posterior striatum.MUSmuscimolSEMStandard error of meanDMSOdimethyl sulfoxide

## Introduction

In everyday life, sounds often predict forthcoming events, allowing for planning and execution of appropriate behavioral responses. Consider, for instance, the confidence with which we step out of an elevator's open doors a few seconds after its chime, even without looking up from our phone. Or how the conclusion of a friend's sentence determines the opportune moment for our response [1–3]. Likewise, animals rely on sound cues to gauge how swiftly they should vocalize in response to a conspecific call [4] evade a predator [5], hone in on prey [6] or approach a needy offspring [7,8]. These and other examples in everyday life require humans and other animals to utilize sounds to predict and precisely time subsequent salient events and to initiate appropriate behavioral responses within the scale of seconds [9,10].

The ability to use sounds to predict when future events will occur and consequently when to initiate appropriate action relies on a number of neural processing stages [11–13]. First, the sound must be detected, processed, and recognized. Second, the predicted amount of time from the sound to the future event is evaluated. And finally, action is initiated when the elapsed time matches the anticipated appropriate time to act. Neural signatures underlying the first stage of this process, namely sound processing and recognition, have been extensively identified in the auditory pathway, and in particular in the auditory cortex (For example: [14–21]). Considerably less is known about the second stage, and specifically where and how the brain encodes the predicted amount of time from the sound to a future event. Traditional models of time perception have suggested the existence of a "centralized clock" in the brain (also referred to as the "Internal clock model") [22–25]. According to this model, the role of sensory regions is to detect the relevant sensory stimulus and communicate this information to higher-order centralized-clock brain regions, where a continuous representation of elapsed time is maintained [11,22,24,25]. This model proposes that the centralized clock is similarly able to estimate time based on cues from varying modalities, arriving via distinct pathways from the various sensory regions and hence, is "amodal" [12,13]. Different studies have implicated a number of brain regions hosting such centralized clocks, including the nucleus accumbens [26], caudoputamen [27], ventral tegmental area [28], substantia nigra compacta [29], dorsal striatum [29–31] and the medial prefrontal cortex [31–33].

However, the universality of this model has been challenged by studies showing that in addition to a centralized clock, there exist sensory-specific timing mechanisms, and that these are located within early sensory cortical regions [11–13]. Early support for this suggestion came from studies demonstrating that the ability to estimate time from a sensory cue depends on the modality of that cue (For example [34,35]). Furthermore, recent studies have identified signatures of time estimation within sensory cortices. For example, in vivo neural recordings in the primary visual cortex of rodents show various neural response forms which represent the time interval between the visual stimulus and the anticipated reward [35–37]. A recent study further showed that this reward timing representation is modulated by an intracortical network of inhibitory interneurons in the visual cortex [38].

In the auditory pathway, a candidate brain region for encoding sound-triggered timing is the auditory cortex (AC) due to its established role in behavior- and decision-making- dependent sound processing [17,39–48]. A large body of studies has demonstrated retrospective coding of the degree to which a sound deviates from expectation in AC [49–55]. However, much less is known about the existence of prospective coding of anticipated time from a sound to a subsequent event. In one study, auditory cortical responses to a tone series varied depending on whether a subsequent target sound was expected early (300–450 ms) or late (1,300–1,500 ms) within the series [56]. However, this study did not test whether AC encodes expectation of upcoming non-auditory cues or whether a continuous representation of predicted time is encoded in AC. In a recent study, mice were trained on a self-paced action timing task, in which lever pressing caused reward delivery after 30 s. Optogenetic stimulation and inactivation of the secondary AC implicated its responses to the sound of lever press as being causally involved in timing reward-preparatory action [57]. However, this study did not test whether and how AC is involved in encoding varying sound-reward time intervals, in particular in the behaviorally-critical timescale of a few seconds. Moreover, the sound in this study was self-generated (via a lever press), which engages distinct neural processing mechanisms [45,58–62]. Thus, it remains unclear whether and how AC is involved in sound-triggered predictive timing of future salient events on the timescale of seconds.

The final step of sound-dependent prediction of imminent events requires initiation of appropriate action once the anticipated time from sound to action has elapsed. To carry this out, the neural information assimilated in the first two stages of this process needs to be sent to downstream brain regions to induce consequent action timing and initiation. Previous studies have implicated a number of brain regions that mediate sound-triggered action initiation, including the dorsolateral striatum [27,32,63,64], the medial prefrontal cortex [32,65] and the supplementary motor area [13,66]. Anatomically, the most prominent candidate brain region to receive sound-triggered time keeping signals from AC and participate in converting this information to action, is the posterior tail of the dorsal striatum (hereafter referred to as the posterior striatum). The posterior striatum (pStr) receives monosynaptic projections from AC [67–69] and these projections are known to be causally involved in auditory-guided decision-making tasks [69] and auditory associative learning tasks [68]. Moreover, pStr itself responds to auditory stimulation [63,70] and has been shown to play a key role in various stimulus-driven time keeping behaviors [27,32]. However, whether and how information about sound-triggered time interval estimation arriving from AC engages pStr remains unknown. To collectively address these gaps, we investigated the causal and functional role of the auditory cortical-striatal circuit in sound-triggered prediction of time to consequent reward.

## Results

### Mice use sound to predict reward time with 1-s temporal resolution

To investigate the role of auditory cortical-striatal neural mechanisms underlying sound-triggered reward timing prediction, we employed an appetitive sound-guided trace conditioning task in water-restricted mice. Following habituation to head fixation, 8 mice underwent training sessions of 150–200 trials each, in which each trial was initiated with a 1.5 s long sound stimulus (composed of a sequence of three pure tones), followed by a fixed delay period and then delivery of water reward (Fig 1A). Trials were separated by inter-trial intervals which randomly varied between 2 - 6 s in duration. All animals started behavioral training sessions with a fixed delay period of 1.5 s between the sound cue termination and reward ("sound-reward time interval"). After 7–10 days of training, we introduced randomly interspersed 10%–20% catch trials, in which the reward was withheld. Mice expressed learning of the sound-reward contingency by increasing their lick rate before the anticipated reward time ("Predictive licking", Fig 1B, left) and by licking before and during the time of anticipated reward in the catch trials (Fig 1B, right).

To determine whether predictive licking reflects a reliable estimation of anticipated reward time, the same mice then went on to train on the same paradigm but with different sound-reward time intervals. We posited that if mice can reliably estimate time on the scale of seconds and use these estimates to guide their behavior, the timing of predictive licking would vary with the duration of the sound-reward interval. To test this, once animals showed reliable performance on

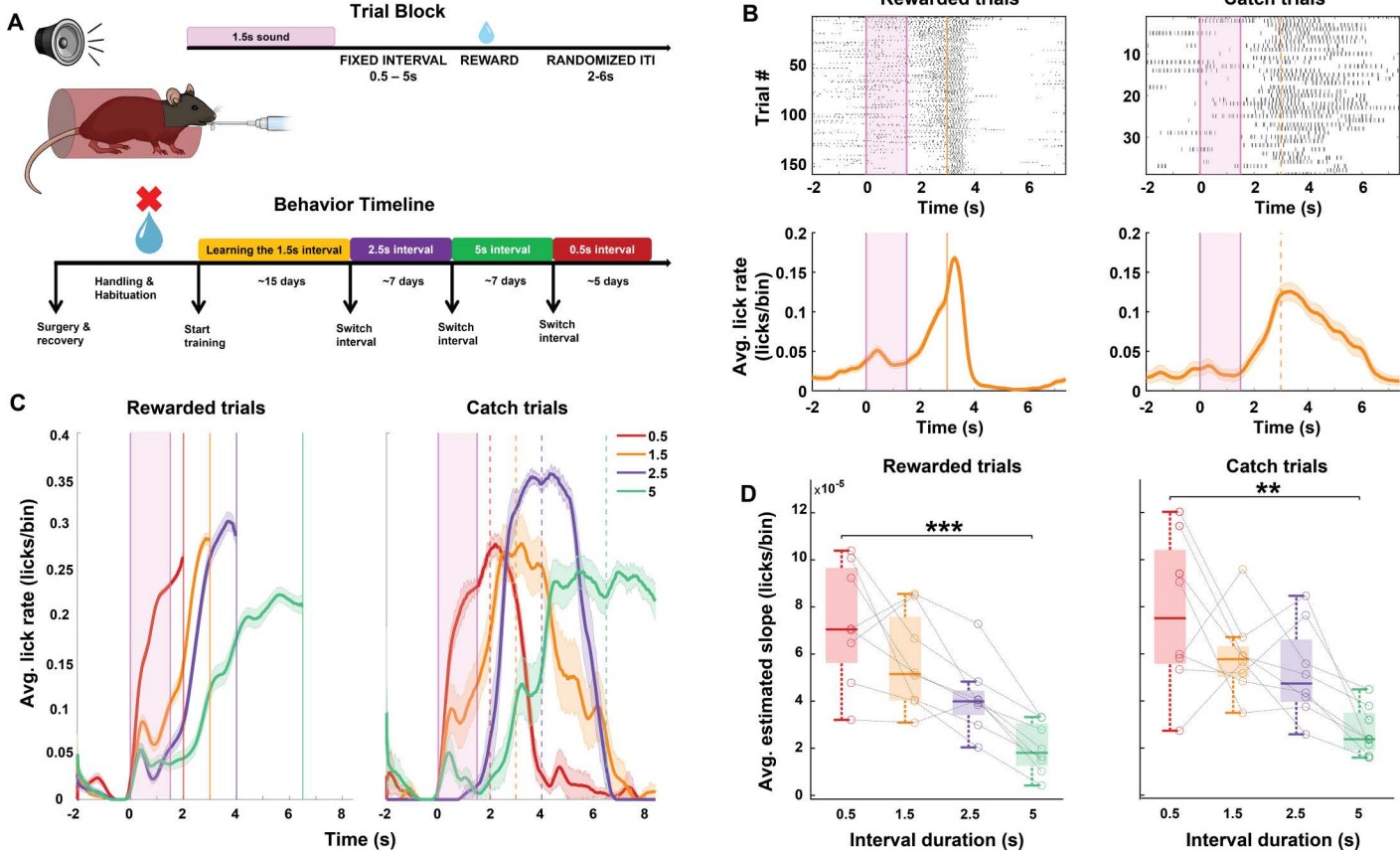

**Fig 1. Mice predict reward timing using a sound cue. A.** An illustration of the behavioral setup for sound-triggered reward time prediction task, components of a trial block and the experimental timeline for behavioral training. **B.** Top: Peri-sound lick raster of an example behavioral session from a trained animal performing on rewarded (left) and catch (right) trials within the session. Bottom: Average peri-sound lick rate response curve (solid line denotes mean, shaded area represents SEM across trials) for the example behavioral session above for rewarded (left) and catch (right) trials. Shaded pink region represents the 1.5 s long sound period. Solid and dotted orange lines represent reward delivery in rewarded trials and expected reward timing in catch trials. Black ticks represent licks. **C.** Average peri-sound lick rate (licks/bin, bin = 1 ms) response curves (solid line denotes mean, shaded area represents SEM across trials) of an example animal trained to perform on the four different sound-reward intervals represented by different colors. Left: Rewarded trials; Right: Catch trials. Shaded pink region represents the 1.5 s long sound period. Solid and dotted lines represent when reward was given in rewarded trials and expected in catch trials for each of the sound-reward interval. **D.** Box plot representing the estimated slope of predictive licking response curves for each of the four sound-reward intervals across all animals (N = 8) for rewarded trials (left, ***p = 0.000389, Kruskal–Wallis test) and for catch trials (right, **p = 0.0011, Kruskal–Wallis test). In each box, the solid line indicates the median, and the bottom and top edges of the box indicate the 25th and 75th percentiles, respectively. The whiskers extend to the minimum and maximum values in the dataset. Individual circles represent the average slope for each animal and per animal data points across sound-reward intervals are connected by the black dashed lines. The data underlying this figure can be found at doi: 10.6084/m9.figshare.29033654.

the task with the 1.5 s sound-reward interval (as evidenced by reliable predictive licking) they relearned versions of this task with a 2.5 s, 5 s and 0.5 s sound-reward time intervals, in this order (Fig 1A, Behavior Timeline). Licking during the baseline period did not change across sound-reward intervals (S1A Fig). For each animal, we compared the slopes of their predictive licking curves on their "best" behavior days of each of these sound-reward time intervals (see "Materials and methods"). We computed the slopes of the predictive licking curves for rewarded and catch trials separately (see "Materials and methods"). We found that the predictive licking slopes varied with the duration of the sound-reward interval (Figs 1C and S2). Across animals, the average slope of the predictive licking curves significantly varied as a function of the sound-reward interval duration, with the shortest interval duration (0.5 s) inducing the steepest slope of predictive

licking, followed by 1.5 s, 2.5 s and 5 s (Fig 1D, Kruskal–Wallis test for multiple comparisons, $p$ = 0.0011 for rewarded trials, $p$ = 0.000389 for catch trials). Similar results were also observed when calculating slopes from 4 alternative data inclusion criteria of days and trials, demonstrating the robustness of these findings (S3 Fig). These results show that mice can estimate time intervals on the scale of 0.5–5 s with at least 1-s temporal resolution and use these estimates to predict the time of expected reward following a sound.

## The auditory cortex is required for sound-triggered delayed reward prediction

The auditory cortex (AC) plays an important role in predictive and behavior- dependent sound processing [41,42,55,71–74]. Since our task requires mice to use the sound to time their behavior, we hypothesized that the AC is necessary for successful performance of this task. To test this hypothesis, we measured the influence of bilateral AC inactivation using the GABA-A receptor agonist muscimol on trained animals' ability to predict reward at a 1.5 s sound-reward time interval. Muscimol expression was histologically validated at the end of each experiment and only animals with selective targeting in AC were included (Figs 2A and S4A–4B). AC inactivation resulted in an overall reduction in the ability to predictively lick for reward as compared to infusion of inert PBS as a control (Fig 2B). An impaired ability to predictively lick for reward following muscimol infusion was consistently observed in all our animals ($N$ = 8, $p < 0.05$ for each mouse, Wilcoxon rank-sum test). To rule out the possibility that reduced predictive licking reflects a simple impairment in sound processing, we trained another cohort of mice on a variation of the task, in which reward was delivered immediately following sound onset (No-Delay Task). Mice trained on the No-Delay task showed evidence for sound-reward association by consistently licking in response to the sound during catch trials, when no reward was delivered (Fig 2C, right). Interestingly, this form of predictive licking following the No-Delay sound-reward association was unaffected by AC inactivation ($N$ = 8, $p > 0.05$ for each mouse, Wilcoxon rank-sum test; Fig 2C). To directly compare the influence of AC inactivation on prediction of delayed and immediate reward, we calculated the log of the ratio of predictive licking in PBS and muscimol ("log PLR") for each of the task versions. Thus, larger log PLR values indicate a greater influence of AC inactivation on predictive licking. Using this metric, we found that AC inactivation had a significantly more detrimental effect on sound-guided prediction of 1.5 s-delayed reward than on immediate reward ($p$ = 0.00031, Wilcoxon rank-sum test, Fig 2D). AC inactivation did not differentially change baseline lick rates in the delayed and No Delay conditions (S1B Fig). These findings demonstrate that the AC is causally involved in sound-triggered time-delayed reward prediction.

## Mice predict reward timing from sound onset

Having established that mice can use the sound cue to predict the timing of reward across varying sound-reward time intervals and that this behavior causally involves the auditory cortex, we next asked whether this behavior reflects time estimation from sound onset or from sound termination (Fig 3A). To this end, we compared the predictive lick pattern under the standard paradigm of a 1.5 s-long sound and 1.5 s-long sound-reward interval, to a similar paradigm in which the sound duration was cropped by 0.5 s, to a duration of 1 s. We argued that if mice use sound onset to predict reward timing, their predictive lick pattern would be unaffected by a shorter sound duration, whereas if mice use sound termination to predict reward timing, predictive licking will start earlier in the trials with the shorter sound cue. To test this, we trained a cohort of 8 mice to predict reward at a 1.5 s sound-reward interval and then introduced them to a training session with 35% short sound trials (Fig 3A). All our animals showed similar behavioral responses on short sound and standard sound trials, with almost identical predictive licking curves (Fig 3B). Across animals, the slopes for the predictive licking curves were not significantly different between the two types of trials (Fig 3C, Wilcoxon rank-sum test, $p$ = 0.96), and the average log PLR for standard sound to short sound trials was not significantly different from 0 (See "Materials and methods", Fig 3D, Wilcoxon signed rank test compared against 0, $p$ = 0.0781). These results suggest that in this task, mice primarily use the sound onset to estimate the amount of time from sound to reward.

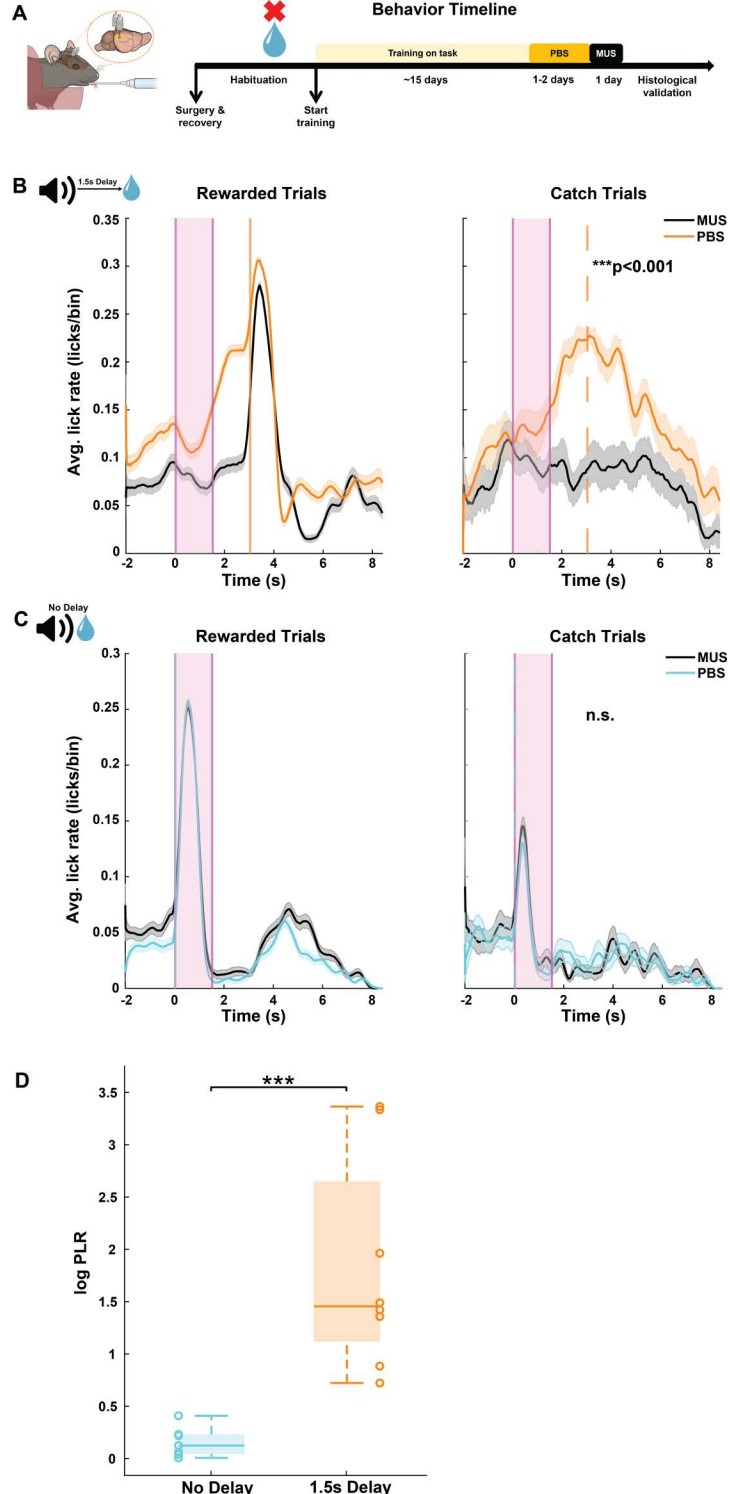

**Fig 2. AC is required for sound-triggered delayed reward prediction. A.** Left: An illustration of the cannula implanted in AC for muscimol infusion. Right: Behavioral timeline for mice that underwent training on the 1.5 s Delay task. **B.** Average peri-sound lick rate (licks/bin, bin = 1 ms) response curves (solid line denotes mean, shaded area represents SEM across trials) of an example animal trained to predict reward at 1.5 s sound-reward interval when infused with PBS (orange) and muscimol (MUS, black) in AC for rewarded trials (left) and catch trials (right). Shaded pink region represents the sound period. Solid and dotted orange lines represent reward delivery in rewarded trials and reward time expectation in catch trials. ***$p$ = 0.0007

(Wilcoxon rank-sum test) denotes the significant difference in predictive licking compared to baseline for catch trials on PBS and MUS conditions. **C.** Average peri-sound lick rate (licks/bin, bin = 1 ms) response curves (solid line denotes mean, shaded area represents SEM across trials) of an example animal trained on the No-Delay task when infused with PBS (light blue) and muscimol (MUS, black) in AC for rewarded trials (left) and catch trials (right). n.s. ($p$ = 0.939, Wilcoxon rank-sum test) denotes the not significant difference in predictive licking between catch trials on PBS and MUS in the No-Delay task. **D.** Significant difference in average log predictive licking ratio (log PLR) between No-Delay ($N$ = 8) and 1.5 s Delay ($N$ = 8) cohorts (\*\*\*$p$ = 0.00031, Wilcoxon rank-sum test).

$$\log Predictive\ licking\ ratio\ (PLR) = \log \frac{Avg.\ across\ trials\ [\#\ of\ licks\ in\ (Predictive\ lick\ period - Baseline\ period)]\ on\ PBS\ or\ Saline}{Avg. across\ trials\ [\#\ of\ lick\ in\ (Predictive\ lick\ period - Baseline\ period)]\ on\ MUS\ or\ CNO}$$

*Where, baseline period = sound onset time – 1900 ms to sound onset time – 1150 ms predictive lick period for delay tasks = reward time – 250 ms to reward time + 500 ms predictive lick period for no-delay task = sound onset time + 750 ms.* In each box, the solid line indicates the median, and the bottom and top edges of the box indicate the 25th and 75th percentiles, respectively. The whiskers extend to the minimum and maximum values in the dataset. Open circles indicate the PLR for each animal in the No-Delay and 1.5 s Delay cohorts. The data underlying this figure can be found at doi: 10.6084/m9.figshare.29033654.

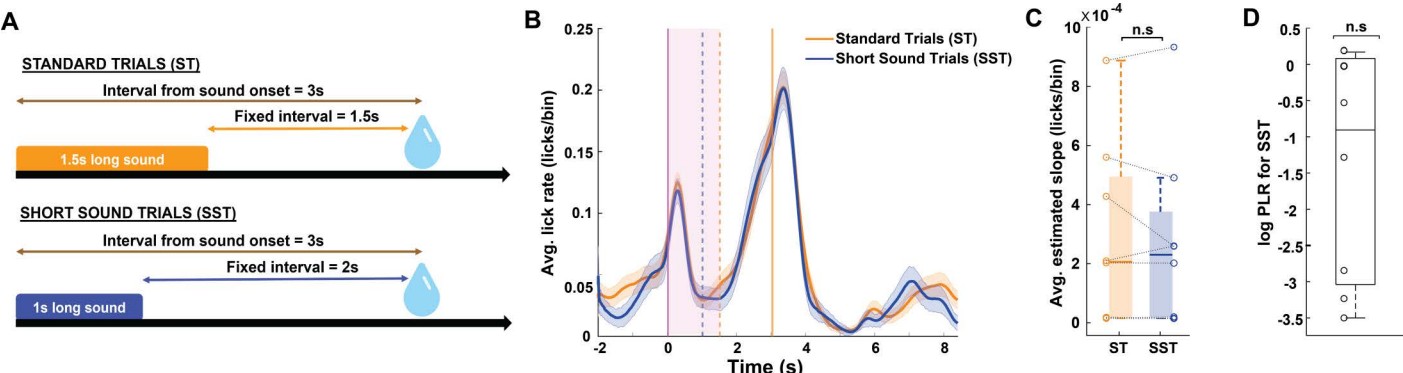

**Fig 3. Mice predict reward timing from sound onset. A.** An illustration of the trial designs. **B.** Average peri-sound lick rate (licks/bin, bin = 1 ms) response curves (solid line denotes mean, shaded area represents SEM across trials) of an example animal trained on the Standard Trials (ST, orange) and Short Sound Trials (SST, navy blue). Dashed lines indicate the sound termination times for ST (orange) and SST (blue) trials. Solid orange line represents when reward was given. Comparison of the predictive licking between ST and SST trials compared to baseline yields $p$ = 0.9 (Wilcoxon rank sum test). **C.** Box plot showing the estimated slope of the predictive licking curves for ST (orange) and SST (navy blue) across animals ($N$ = 8, $p$ = 0.96, Wilcoxon rank-sum test). In each box, the solid line indicates the median, and the bottom and top edges of the box indicate the 25th and 75th percentiles, respectively. The whiskers extend to the minimum and maximum values in the dataset. Lines connecting open circles represent the estimated slope values for each animal in ST and SST. **D.** Average log predictive licking ratio (log PLR for SST) across animals is not significantly different from 0 ($N$ = 8, $p$ = 0.0781, Wilcoxon signed rank test compared against 0). In each box, the solid line indicates the median, and the bottom and top edges of the box indicate the 25th and 75th percentiles, respectively. The whiskers extend to the minimum and maximum values in the dataset. Open circles represent the log PLR for each animal. Log PLR is calculated as in Fig 2. The data underlying this figure can be found at doi: 10.6084/m9.figshare.29033654.

### AC sound responses encode predicted time-to-reward

A prominent model for the neural processing underlying sound-triggered time predictions suggests that following sound coding in the auditory pathway, downstream brain regions encode the anticipated time interval from the sound to subsequent events [11,22,24,25]. Alternatively, given the established role of the AC in predictive coding, the predicted time from sound to reward may already be reflected in the AC response to the sound itself. To test this possibility, we recorded local field potential (LFP) activity in the AC of the right hemisphere in 8 mice, as they were trained to predict timed reward using sound at the four different sound-reward intervals in the order described previously. We histologically verified the position of these electrodes at the end of the experiments (Fig 4A). Similar to our findings in the first cohort (Fig 1), animals in this

cohort also changed the rate of their predictive licking curves according to the duration of the sound-reward intervals, with the slope decreasing with increased interval durations for both rewarded and catch trials (Fig 4B–4C, $p$ = 0.000043 for rewarded trials and $p$ = 0.00008 for catch trials; Kruskal–Wallis test for group differences). We similarly identified the best behavior performance day of each sound-reward interval for analysis and excluded trials in which the animal moved or in which the animal licked in the 200 ms period from sound onset to avoid a contribution of motor activity to the sound response magnitude [45,58,75]. We then quantified the magnitude of the AC responses to the sound for each of the interval durations per animal. As our behavioral results showed that mice use the sound onset to estimate the time to reward, we focused on the responses to sound onset. Interestingly, we found that AC responses (to the same sound) increased in magnitude with increasing sound-reward intervals (Fig 4D). Across animals, the average normalized sound response magnitude was significantly different across interval durations ($p$ = 1.426 × 10$^{-36}$, Kruskal–Wallis test), with the response magnitude increasing as a function of the time interval from sound to reward (Fig 4F). Similar results were also observed when calculating response magnitudes using 4 alternative data inclusion criteria of days and trials, demonstrating the robustness of these findings (S3 Fig). As our sound was a sequence of 3 tones, we also quantified responses to the second and third tones for the different interval durations and found that they were considerably smaller than responses to the first tone and did not scale with interval duration (S5 Fig). We also quantified the magnitude of responses to the sound offset for each of the interval durations per animal and found that offset responses did not significantly change in magnitude across sound-reward intervals ($p$ = 0.263, Kruskal–Wallis test, Fig 4E and 4G). Together, these results indicate that in this task, AC responses to the sound onset encodes, beyond the sound itself, the predicted time from sound to reward.

A possible alternative interpretation to these findings is that they reflect AC encoding of sound valence rather than sound-reward interval duration. Later rewards could have subjectively lower value due to "delayed reward discounting" [76], and since AC sound responses can be modulated by the associated reward [70,77–79], the graded AC response magnitudes to varying sound-reward intervals could in principle encode sound valence. To test this possibility, we conducted a different series of experiments, in which we changed the reward size while keeping the reward timing constant (S6 Fig). We found that post-learning behavioral performance did not significantly differ across the three reward sizes (S6B–S6C Fig). Further, AC response magnitude to the sound onset did not significantly differ across these three reward sizes (S6D–S6E Fig). These results suggest that the AC LFP onset response magnitude to varying sound-reward intervals encodes the interval duration rather than delay-discounted reward value.

### Activity of auditory cortical neurons that project to posterior striatum is necessary for sound-guided prediction of delayed reward

For successful sound-triggered reward timing prediction, animals need to translate their prediction of reward time into motor action, which in the current task is licking. While our data suggests that the AC is involved in sound-triggered reward timing prediction, we hypothesized that this behavior and its translation into motor action would further depend on the posterior striatum (pStr). The pStr is a key candidate brain region to support this function, as it receives strong monosynaptic projections from the AC [67–69] as well as from the thalamic medial geniculate body [80–82] and is involved in sound processing and sound-guided behaviors [68,69]. Moreover, pStr itself is known to be involved in appetitive auditory frequency discrimination tasks [63,70]. Hence, we investigated the role of auditory cortical projections to the pStr in sound-triggered reward time prediction.

To address this, we chemogenetically inactivated pStr-projecting AC neurons in animals trained to predict timed reward using a sound cue. Using a dual virus approach, we bilaterally injected and expressed a Cre-dependent DREADD virus (AAV-hSyn-DIO-hM4D(Gi)-mCherry) in the AC of mice in the experimental group, (or AAV-hSyn-DIO-mCherry in the control group), and a retrograde-CRE virus (pENN/AAVrg-hSyn-Cre-WPRE-hGH) in the pStr. This approach allowed us to target the projections from the AC to pStr for chemogenetic inactivation, although sparse collateral projections may have also been targeted [5] (see "Discussion"). The expression of these viruses was histologically validated at the end of the

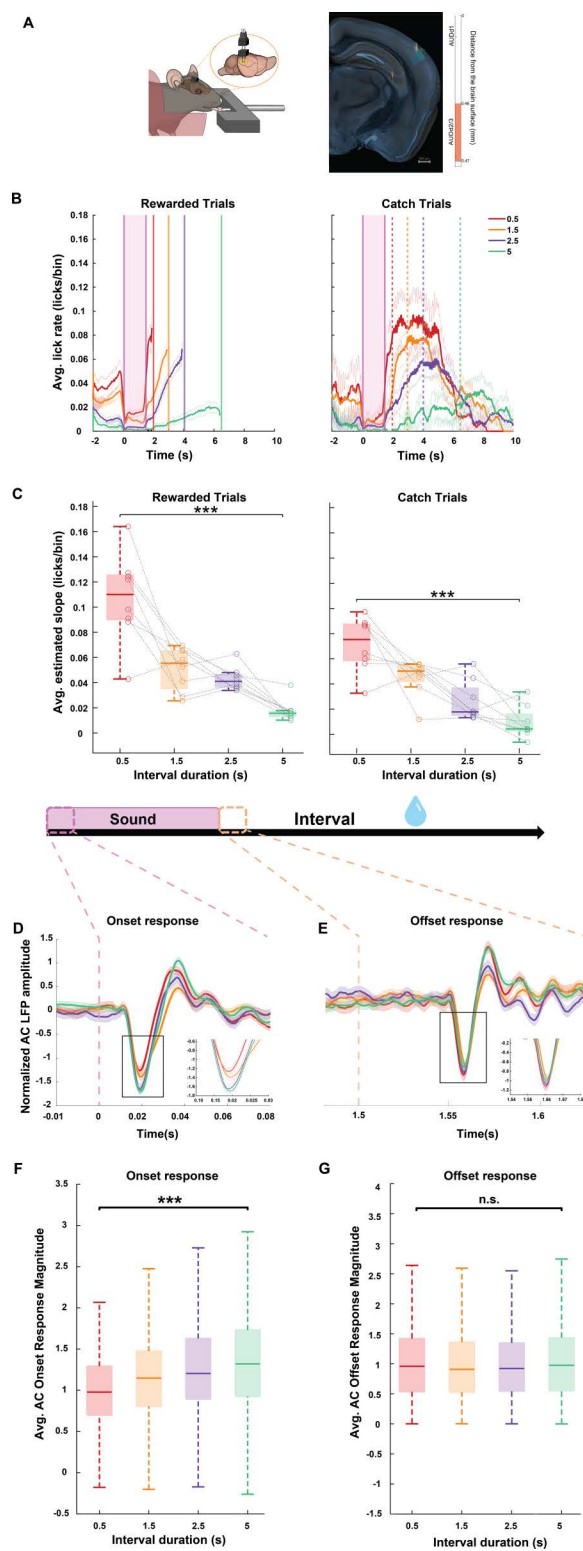

**Fig 4. Auditory cortical sound responses encode predicted time to reward. A.** Left: An illustration of the electrodes implanted in right AC. Right: Histological verification of electrode position in AC of the right hemisphere. The brain slice acquired from an example animal is overlaid with the corresponding coronal section from the Allen Mouse Brain Atlas (see "Materials and methods"). The electrode track is indicated by the green dotted line

on the brain slice and the markers on the right indicate the depth at which the electrode was implanted in the right hemisphere. Scale bar: 500 μm. **B.** Average peri-sound lick rate (licks/bin, bin = 1/20 ms) response curves (solid line denotes mean, shaded area represents SEM across trials) of an example animal trained to perform on the four different sound-reward intervals represented by different colors. Left: Rewarded trials; Right: Catch trials. Shaded pink region represents the 1.5 s long sound period. Solid and dotted lines represent when reward was given in rewarded trials and expected in catch trials for each of the sound-reward interval. **C.** Box plot representing the estimated slope of predictive licking response curves for each of the four sound-reward intervals across all animals ($N = 8$) for rewarded trials (left,***$p = 0.000043$, Kruskal–Wallis test) and for catch trials (right, ***$p = 0.00008$, Kruskal–Wallis test). In each box, the solid line indicates the median, and the bottom and top edges of the box indicate the 25th and 75th percentiles, respectively. The whiskers extend to the minimum and maximum values in the dataset. Individual circles represent the average slope for each animal and per animal data points across sound-reward intervals are connected by the black dashed lines. **D.** Normalized average AC LFP responses to the sound onset from an example animal trained on the four different sound-reward intervals represented by the different colors (solid line denotes mean, shaded area represents SEM across no-lick trials). Dashed pink lines represent the period from sound onset as shown in the illustration above. Inset: Magnified AC LFP responses to the sound onset for the region indicated by the black rectangle. **E.** Normalized average AC LFP responses to the sound offset from an example animal trained on the four different sound-reward intervals represented by the different colors (solid line denotes mean, shaded area represents SEM across no-lick trials). Dashed yellow lines represent the period from sound offet as shown in the illustration above. Inset: Magnified AC LFP responses to the sound offset for the region indicated by the black rectangle. **F.** Boxplot showing the onset response magnitudes computed across animals ($N = 8$) for each of the sound-reward intervals. In each box, the solid line indicates the median, and the bottom and top edges of the box indicate the 25th and 75th percentiles, respectively. The whiskers extend to the minimum and maximum values in the dataset. Comparison across sound-reward intervals yields ***$p = 1.426 \times 10^{-36}$ (Kruskal–Wallis test). **G.** Boxplot showing the offset response magnitudes computed across animals ($N = 8$) for each of the sound-reward intervals. In each box, the solid line indicates the median, and the bottom and top edges of the box indicate the 25th and 75th percentiles, respectively. The whiskers extend to the minimum and maximum values in the dataset. Comparison across sound-reward intervals yields $p = 0.263$ (Kruskal–Wallis test). The data underlying this figure can be found at doi: 10.6084/m9.figshare.29033654.

experiments and only those with targeted virus expression in AC cell bodies and pStr axonal projections were included (Fig 5B).

Mice were trained to reliably predict sound-guided reward at a 1.5 s sound-reward interval. They were then injected with saline (i.p.) as a control and their behavioral performance was recorded 30 min after the injection. Twenty-four hours later, we injected them with CNO (5 mg/kg, i.p.) and after 30 min, again recorded their behavioral performance (Fig 5A–5B). Chemogenetic silencing of the AC-pStr projections in the mice of the experimental group significantly reduced their ability to predict reward at 1.5 s (Fig 5C). We observed this significant effect in all mice in the experimental group ($p < 0.05$ for each mouse, Wilcoxon rank-sum test). In contrast, none of the mice in the control group showed a significant change in predictive licking following CNO injection compared to their performance with saline injection (Fig 5D, $p > 0.05$ for each mouse, Wilcoxon rank-sum test). These findings demonstrate that the targeted AC-pStr projection is necessary for sound-triggered prediction of delayed reward. To verify that chemogenetic inactivation of AC-pStr projections does not impair animals' ability to process sounds and form sound-reward associations, we trained all mice in the experimental and control groups on the No-Delay task following another 4-day CNO washout period and followed the same manipulation protocol above (Fig 5A). Mice in both these groups did not show an impairment in their ability to lick for reward immediately following sound in the catch trials (Fig 5E–5F).

At a population-level, we quantified the effect of the AC-pStr projection inactivation on behavior using the log PLR for Saline to CNO days (see "Materials and methods"), such that higher values indicate a reduction in animals' ability to reliably predict the time-to-reward from sound. We found that the average log PLR was significantly higher for the 1.5 s sound-reward delay compared to the No-Delay (Fig 5G, $p = 0.02$, Wilcoxon rank-sum test). In contrast, mice in the control group showed no significant difference in the average log PLR across No-Delay and 1.5 s (Fig 5H, $p = 0.366$, Kruskal–Wallis test). These findings further establish the causal involvement of the AC-pStr projections in animals' ability to predict delayed reward using a sound cue.

To investigate the functional consequence of inactivation of the AC-pStr pathway on pStr activity, in a subset of 3 animals from the experimental group, we acquired LFP responses in pStr as these animals trained on the 1.5 s-Delay and No-Delay tasks (S7 Fig). We filtered for no-lick trials and computed the average response magnitude of pStr responses at sound onset. pStr sound onset response magnitude significantly reduced in all the 3 animals from saline to CNO days for the 1.5 s-Delay task (S7B and S7D Fig, $p < 0.05$ for each mouse, Wilcoxon rank-sum test). We did not observe a

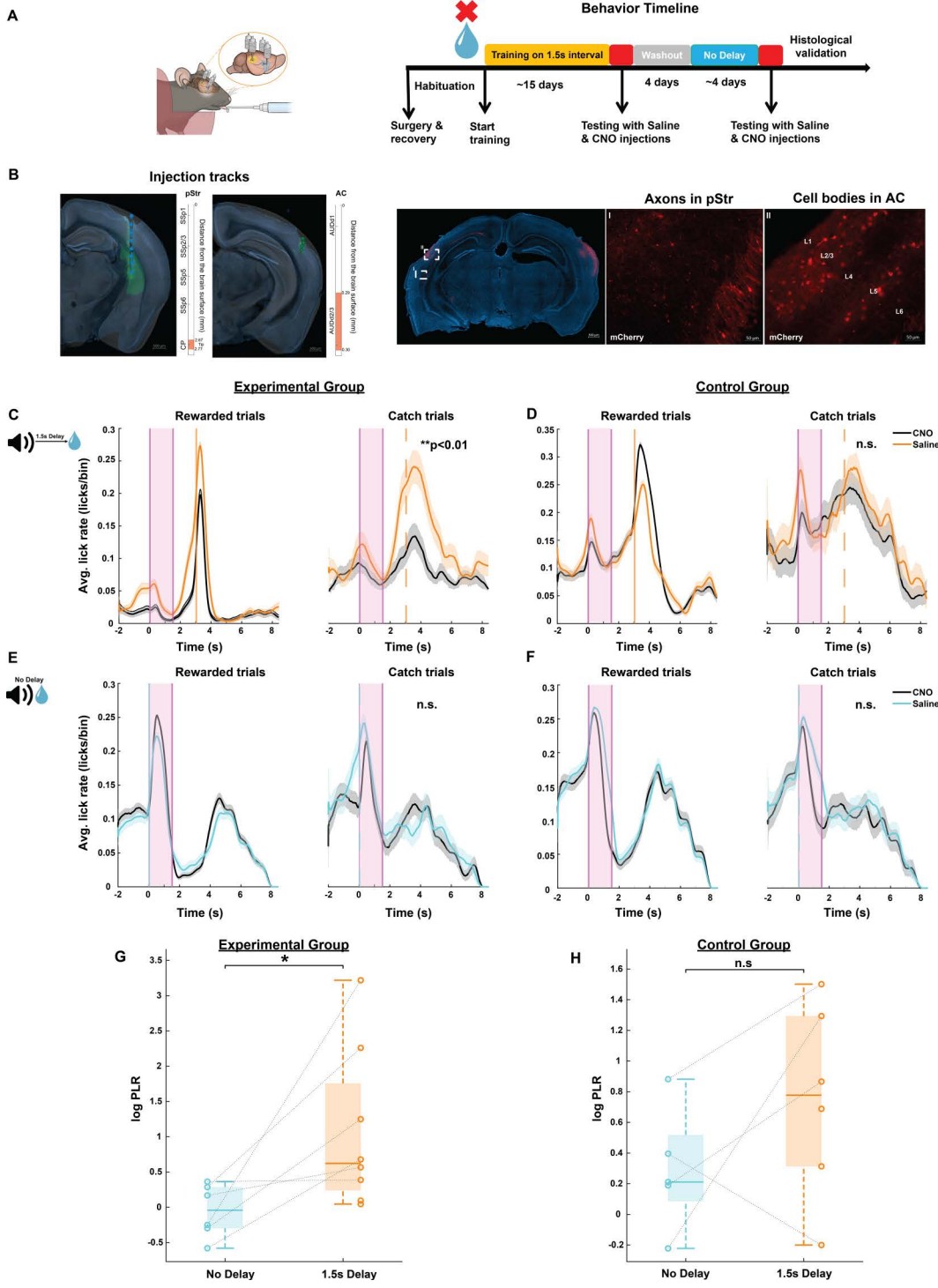

**Fig 5. AC to pStr projections are causally involved in sound-triggered delayed reward prediction. A.** Left: An illustration of cannulas implanted in bilateral AC and pStr for virus injections. Right: Behavioral timeline for chemogenetic inactivation experiments. **B.** Histological verification of selective virus expression in the projections from AC to pStr. A brain slice acquired from an example animal is overlaid with the corresponding coronal section from the Allen Mouse Brain Atlas for both and AC and pStr (see "Materials and methods"). Left: Virus injection tracks in pStr and AC denoted by green dotted line on the brain slice and the markers on the right indicate the depth at which the electrode was implanted in the right hemisphere. Scale bar: 500 µm Right: Magnified images of DREADD virus expression in pStr axons (I) and AC cell bodies **(II)**. Scale bar: 50 µm. **C–F.** Average peri-sound lick rate (licks/

bin, bin = 1 ms) response curves (solid line denotes mean, shaded area represents SEM across trials) of example animals trained to predict reward at either 1.5 s sound-reward interval **(C and D)** or at no delay **(E and F)** when injected with saline (orange or light blue) and CNO (black) for rewarded trials (left) and catch trials (right). Shaded pink region represents the 1.5 s long sound period. Solid and dotted orange or light blue lines represent when reward was given in rewarded trials and expected in catch trials. Left column represents animals from the experimental group and right column represents animals from the control group. **G and H**. Box plot showing the log Predictive Licking Ratio (log PLR) across animals trained on the 1.5 s Delay and No-Delay tasks in the experimental group (left, *N* = 8) and in the control group (right, *N* = 6). Lines connecting the circles represent the log PLR for each animal when trained on the 1.5 s Delay and No-Delay tasks. The average log PLR for experimental group animals was significantly higher for 1.5 s Delay task than No-Delay task (*\*p* = 0.02, Wilcoxon rank-sum test) and was not significantly different for the control group animals (*p* = 0.366, Wilcoxon rank-sum test). In each box, the solid line indicates the median, and the bottom and top edges of the box indicate the 25th and 75th percentiles, respectively. The whiskers extend to the minimum and maximum values in the dataset. Log PLR is calculated as in Figs 2 and 3. The data underlying this figure can be found at doi: 10.6084/m9.figshare.29033654.

significant change in pStr onset response magnitude from saline to CNO days for the No-Delay task (S7C and S7E Fig, *p* > 0.05 for each mouse, Wilcoxon rank-sum test). This selective reduction in pStr LFP response magnitude following the chemogenetic inactivation of the AC-pStr projections in the 1.5 s-Delay task provides neurophysiological corroboration of the causal involvement of these AC-pStr projections in prediction of reward timing using a sound cue. Chemogenetic inactivation did not differentially change baseline lick rates in the delayed and No Delay conditions (S1D–S1E Fig). Together, these findings show that AC-pStr projections are a necessary neural pathway for sound-guided delayed reward prediction behavior.

### The posterior striatum is involved in sound-guided reward time prediction

We next asked whether the target region of these AC-pStr projections, pStr itself, is a necessary component of the neural circuitry supporting sound-guided reward prediction behavior. We implanted mice with bilateral cannulae in pStr (S4C–S4D Fig) and compared their ability to predict reward at 1.5 s with muscimol infusion to that with PBS infusion on a day prior. We found that inactivation of pStr reduced animals' ability to predict the time-delayed reward (Fig 6A, *p* < 0.05, Wilcoxon rank-sum test). Similar to the previous inactivation experiments, we trained another cohort of mice on the No-Delay task and compared their behavioral performances following PBS and muscimol infusion into pStr. Inactivation of pStr using muscimol did not change animals' behavior on the No-Delay task (Fig 6B, *p* > 0.05, Wilcoxon rank-sum test), confirming that there was no impairment in animals' ability to process sounds or lick for reward. Across animals, the average log PLR for PBS to muscimol days for the 1.5 s Delay task was significantly higher than that of the No-Delay task (Fig 6C, *p* = 0.0079, Wilcoxon rank-sum test). pStr inactivation did not differentially change baseline lick rates in the delayed and No Delay conditions (S1C Fig). Overall, these findings show that pStr is also required for successful sound-guided reward prediction behavior.

### Coordination of sound-evoked responses in AC and pStr during sound-triggered reward time prediction

To determine the activity patterns in pStr and to test whether the AC and pStr activity is coordinated during sound-triggered reward time prediction behavior, we simultaneously recorded LFP activity in AC and pStr as animals learned to predict sound-triggered reward timing at the four sound-reward time intervals (Fig 7A). Like we did for analyzing AC responses, we eliminated trials with movement and included only trials with no licks in the first 200 ms from sound onset for the best behavior days on each sound-reward interval. As expected from previous studies [70,83], we found robust sound responses in pStr, albeit of lower magnitude than in the AC. When we compared the average pStr sound responses across all four sound-reward intervals, we found that pStr responses also tended to increase with sound-reward interval duration (Fig 7B). However, at the population level, we noticed that while the pStr response magnitude was significantly different across sound-reward intervals (Fig 7C, $p = 7.18 \times 10^{-10}$, Kruskal–Wallis test), the pair-wise comparison of the response magnitude across all pairs of sound-reward intervals did not yield significant differences, unlike responses in the AC.

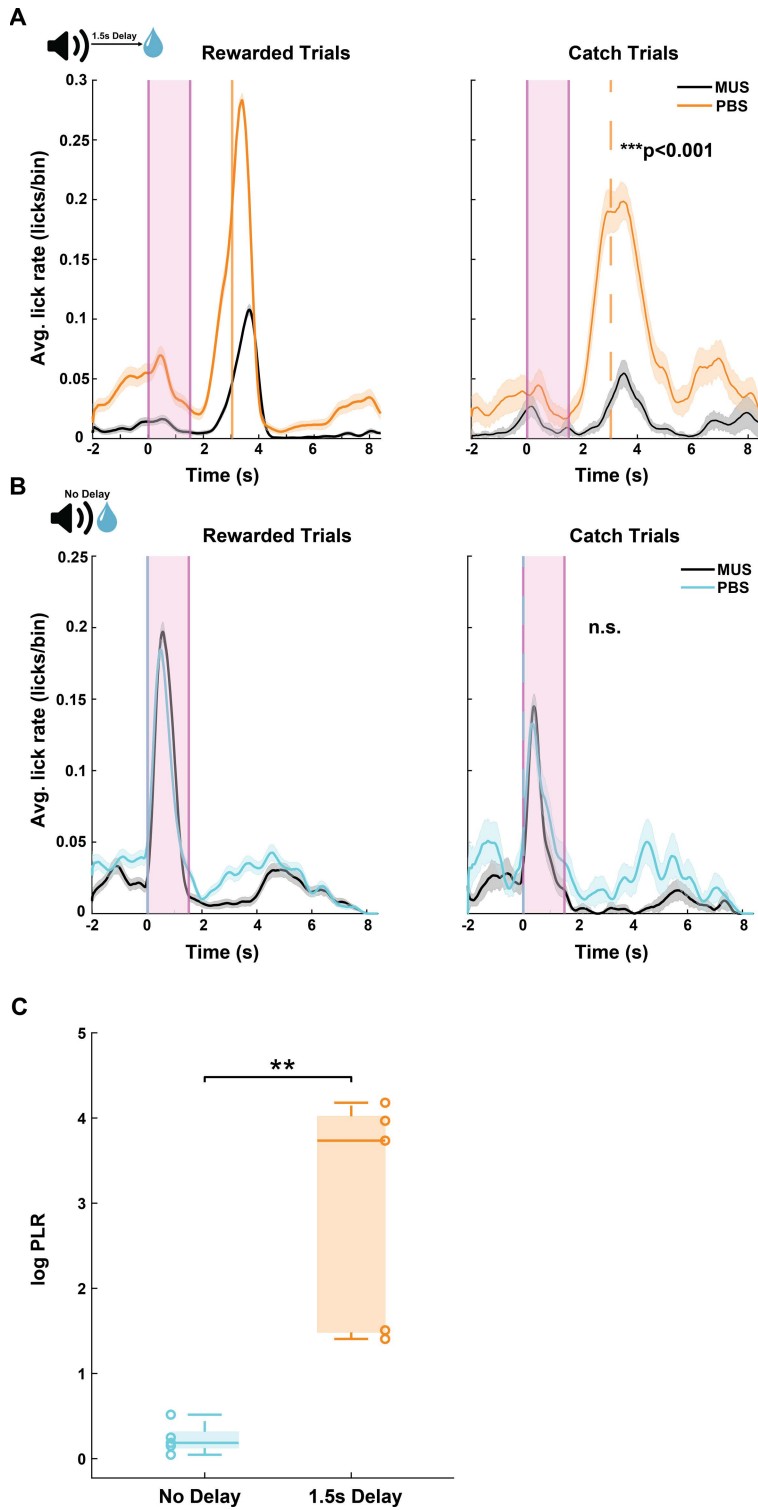

**Fig 6. pStr is causally involved in sound-triggered delayed reward prediction. A**. Average peri-sound lick rate (licks/bin, bin = 1 ms) response curves (solid line denotes mean, shaded area represents SEM across trials) of an example animal trained to predict reward at 1.5 s sound-reward interval when infused with PBS (orange) and muscimol (MUS, black) in pStr for rewarded trials (left) and catch trials (right). Shaded pink region represents the 1.5 s long sound period. Solid and dotted orange lines represent when reward was given in rewarded trials and expected in catch trials. ***$p$ = 0.000007 (Wilcoxon rank-sum test) denotes the significant difference in predictive licking between catch trials on PBS and MUS. **B.** Average peri-sound lick rate (licks/bin, bin = 1 ms) response curves (solid line denotes mean, shaded area represents SEM across trials) of an example animal trained

on the No-Delay task when infused with PBS (light blue) and muscimol (MUS, black) in pStr for rewarded trials (left) and catch trials (right). Shaded pink region represents the 1.5 s long sound period. Solid and dotted blue lines represent when reward was given in rewarded trials and expected in catch trials. n.s. ($p$ = 0.3, Wilcoxon rank-sum test) denotes the not significant difference in predictive licking between catch trials on PBS and MUS in the No-Delay task. **C**. Significant difference in average log predictive licking ratio (log PLR) between No-Delay ($N$ = 5) and 1.5 s Delay ($N$ = 5) cohorts (**$p$ = 0.0079, Wilcoxon rank-sum test). In each box, the solid line indicates the median, and the bottom and top edges of the box indicate the 25th and 75th percentiles, respectively. The whiskers extend to the minimum and maximum values in the dataset. Open circles indicate the PLR for each animal in the No-Delay (blue) and 1.5 s Delay cohorts. Log PLR is calculated as in Figs 2–3, and 5. The data underlying this figure can be found at doi: 10.6084/m9.figshare.29033654.

Next, we asked whether neural activity across the two brain regions was coordinated during sound-guided reward time prediction behavior. We found that AC and pStr LFP response magnitudes showed a significant positive trial-by-trial correlation for each behavioral session (Fig 7D) and the average correlation coefficients across animals did not significantly differ across sound-reward intervals (Fig 7E, $p$ = 0.949, Kruskal–Wallis test). These results suggest the existence of strong coordination across the AC and pStr during sound processing within our task.

pStr receives direct projections from the AC, but both the pStr and AC receive direct projections from the medial geniculate body [80,81,84,85]. We next sought to test to what extent this anatomical projection pattern is reflected in the temporal relationship of AC and pStr sound-evoked responses. To this end, we calculated trial-wise cross-correlations of LFP responses in AC and pStr over a 2 s time period, from 0.5 s prior to sound onset till sound termination (Fig 7F and 7G). The trial-averaged cross-correlation showed a clear peak at 0 lag, indicating strong synchrony in activity across the brain regions, as expected from the shared common input from the MGB (Fig 7G, black trace). However, we noticed that the shape of these average cross-correlation curves was not symmetric around 0. To test this, we compared them to cross-correlations generated after randomly shuffling the AC/pStr identity of the traces (see "Materials and methods"). This form of shuffling generated cross correlations with no temporal directionality across the brain regions by design (Fig 7G, red trace). In comparison to these shuffled cross-correlations, the real cross correlations showed higher values at positive lags, indicating a temporal lead of AC relative to pStr (Fig 7G). To quantify this, we calculated the difference traces of the real and directionality-shuffled cross correlations (Fig 7H). Across the data, the median peak time of these difference traces was 38 ms and was significantly larger than 0 s (Fig 7I, Wilcoxon signed rank test, $p$ = 0.0015). Further, these peak times of difference traces were not significantly different across sound-reward intervals (Fig 7J, $p$ = 0.79, Kruskal–Wallis test). This positive median peak time indicates that in addition to the no-lag synchrony across these brain regions, AC leads pStr during the sound predicting the time to reward in our task.

## Discussion

In this study, we established a novel behavioral paradigm based on an extension of the classical appetitive trace-conditioning task to assess sound-triggered reward time prediction in mice. Using this behavioral paradigm, we found that mice can use a sound cue to reliably predict time intervals to reward at a 1-s temporal resolution and that the ability to predict delayed reward is dependent on AC. We further found that in our behavioral paradigm, mice use the sound onset to estimate the time to reward and that the AC LFP responses to sound onset predicted the amount of time to reward. As a downstream pathway to non-auditory brain areas, we found that the activity of pStr-projecting AC neurons, as well as the pStr itself, are necessary for sound-triggered delayed reward prediction. Sound responses in pStr also varied based on the time to reward. Finally, using simultaneous recordings in AC and pStr during performance of this task, we found strong coordination of neural activity across these brain regions, with synchronous, as well as AC-leading components of cross-correlation. Together, our findings identify AC-pStr mechanisms for sound-triggered prediction of reward timing.

Decades of research have yielded several competing models for how the brain represents time. The most explored theory underlying the timing mechanisms in animals and humans assumes the existence of an internal clock based on neural counting [22–25,86]. According to this theory, higher order "centralized" clock brain regions receive inputs from various

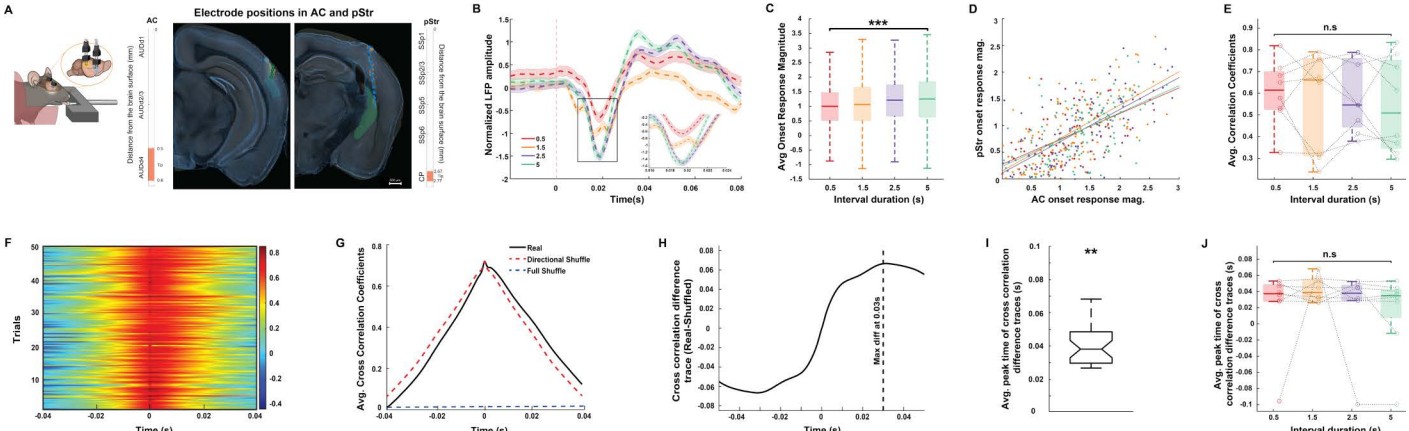

**Fig 7. Coordination of LFP activity across AC and pStr during sound-triggered reward time prediction** **A.** Left: Illustration of electrode implanted in right AC and pStr. Right: Histological verification of electrode positions in AC (left) and pStr (right). The brain slice acquired from an example animal is overlaid with the corresponding coronal section from the Allen Mouse Brain Atlas (see "Materials and methods"). The electrode tracks are indicated by the green dotted line on the brain slices and the markers in the middle indicate the depth at which the electrode was implanted in the right hemisphere. Scale bar: 500 μm. **B.** Normalized average pStr LFP (solid line denotes mean, shaded area represents SEM across no-lick trials) recorded in response to the sound onset from an example animal trained on the four different sound-reward intervals (represented by the different colors). Dashed pink lines represent the period from sound onset as shown in the illustration above. Inset: Magnified pStr LFP responses to the sound onset for the region indicated by the black rectangle. **C.** Boxplot showing the pStr onset response magnitude computed across animals ($N = 8$) for each of the sound-reward intervals. In each box, the solid line indicates the median, and the bottom and top edges of the box indicate the 25th and 75th percentiles, respectively. The whiskers extend to the minimum and maximum values in the dataset. Comparison across sound-reward intervals yields ***$p = 7.18 \times 10^{-10}$ (Kruskal–Wallis test). **D.** Scatter plot of the trial-wise correlation between AC and pStr onset response magnitudes for each of the sound-reward intervals for an example animal. Filled circles of different colors represent the trial-wise onset response magnitudes and the solid lines represent the linear fits for each of the sound-reward intervals. Pearson correlation coefficients for each of the sound-reward intervals was positive and significant: 0.5 s interval (red): $r = 0.682$, $p < 0.001$; 1.5 s interval (orange): $r = 0.754$, $p < 0.001$; 2.5 s interval (purple): $r = 0.788$, $p < 0.001$; 5 s interval (green): $r = 0.613$, $p < 0.001$. **E.** Boxplot showing the distribution of correlation coefficients across animals across sound-reward intervals are not significantly different ($p = 0.949$, Kruskal–Wallis test). In each box, the solid line indicates the median, and the bottom and top edges of the box indicate the 25th and 75th percentiles, respectively. The whiskers extend to the minimum and maximum values in the dataset. Individual circles represent the average value for each animal and per animal data points across sound-reward intervals are connected by the black dashed lines. **F.** Heat map representing the trial-wise cross-correlation of AC and pStr sound-evoked LFP responses for an example animal trained on an 0.5 s sound-reward interval behavioral session. Time axis is set such that it denotes AC leading on the positive side and pStr leading on the negative side. Color bar indicates the cross-correlation coefficients. **G.** Average cross correlation of AC and pStr sound evoked LFP responses across trials for the behavioral session shown above is indicated in black. The dotted red line shows the average of the directionally shuffled cross correlation computed from randomized AC and pStr sound evoked LFP responses from the session. The dotted blue line shows the average of the completely shuffled cross correlation computed from randomized AC and pStr sound evoked LFP responses from the session. **H.** The solid black line shows the difference trace between the real and shuffled average cross correlation in panel G. This cross-correlation difference trace has a peak at 0.03 s as indicated by the dotted black line. **I.** Boxplot showing the peak times of the cross-correlation difference traces computed across all animals and across all sound-reward intervals is significantly greater than 0 s (**$p = 0.0015$, Wilcoxon signed rank test). The solid line indicates the median, and the bottom and top edges of the box indicate the 25th and 75th percentiles, respectively. The whiskers extend to the minimum and maximum values in the dataset. Individual circles represent the average value for each animal and per animal data points across sound-reward intervals are connected by the black dashed lines. **J.** Average peak times of the cross-correlation difference traces across all animals was not significantly different across sound-reward intervals ($p = 0.79$, Kruskal–Wallis test). The central solid line indicates the median, and the bottom and top edges of the box indicate the 25th and 75th percentiles, respectively. The whiskers extend to the minimum and maximum values in the dataset. The data underlying this figure can be found at doi: 10.6084/m9.figshare.29033654.

modalities and maintain a representation of time. On the other hand, parallel work in this domain supports the existence of multiple timing mechanisms distributed across different brain regions and circuits that are engaged based on the task design, sensory modality used, and the temporal resolution of the task [12,13,87,88]. Our findings that sound-triggered reward time prediction ability in mice at a 1-s temporal resolution is dependent on AC, strongly supports the distributed modality-specific timing model. By pharmacologically inactivating AC, we show an impairment in this ability to predict delayed reward time based on a sound cue. Further, we found that the magnitude of AC responses to the sound cue in this task encode

and maintain a neural representation of the time from the sound onset to reward. These results provide evidence for an auditory-modality specific timing mechanism for sound-triggered time estimation to an imminent salient event.

Previous studies have identified different forms of neurophysiological signatures of interval timing across various brain regions [87]. A group of studies in rodent visual cortex [35,37], basal amygdala [89] and basal ganglia [90] have found coding of interval timing via sustained increase or decrease in spiking activity from cue onset till upcoming salient event (reward or foot shock). Other studies, primarily in the prefrontal cortex (PFC) and dorsal striatum, have found gradual firing rate ramping up as the expectation of the animal for an upcoming salient stimulus increases [27,91,92]. Another set of studies have found encoding of time intervals by phasic increases in neural activity at the time of anticipated reward in neurons in the PFC [65], dopaminergic neurons in the ventral tegmental area [28] and in the visual cortex [35]. A common factor across most of these studies is that the amount of time to the salient event is encoded separately from the sensory stimulus that triggers its anticipation. In contrast, we find that the magnitude of the AC responses to the sound itself also encode the anticipated time to reward. Thus, the cue and the reward-timing prediction associated with it are jointly coded in the AC. This joint encoding of the sound cue and the reward-timing in AC follows the findings from a previous study [93], in which single-unit recordings in the AC of mice trained on an aversive auditory trace conditioning task with a 5 s delay between the sound cue and aversive stimulus, show a simultaneous encoding for the sound cue and the reinforcement at sound onset time.

As a potential pathway linking sound and action, we investigated the involvement of auditory corticostriatal projection and found that it is necessary for performance of sound-triggered reward prediction behavior. It should be noted that while our dual-virus targeting approach ensured that only AC-pStr projections were inactivated, some collateral axons from AC to other targets, such as the MGB, superior colliculus or dorsal periaqueductal gray, may have also been inactivated. Although these collaterals have been found to be sparse and therefore unlikely to dominate the behavioral outcome [5], we cannot determine that the AC-pStr projection was the exclusive pathway mediating the behavioral impairment. Nevertheless, the anatomical robustness of the AC-pStr projection, as well as our findings that the inactivation of the AC-pStr pathway induced a significant decrease in the pStr response magnitude to the sound predicting the interval to reward, and that direct inactivation of pStr impaired the animal's ability to predict the sound-delayed reward time, suggest that the AC-pStr projection is involved in mediating this behavior.

The AC projection to pStr encompasses both excitatory and GABAergic inhibitory neurons [5,85,94]. In this study, we cannot conclusively determine whether our findings are primarily mediated by silencing of glutamatergic excitatory neurons or via disinhibition. However, two pieces of evidence support the former alternative. We show that when the pStr-projecting AC neurons are chemogenetically inactivated, there is a significant decrease in pStr sound onset response magnitude (S7 Fig). Further, we show that a similar behavioral impairment as seen following the chemogenetic inactivation is observed following direct pharmacological inactivation of pStr using muscimol, a GABA-A agonist (Fig 6). While further research is required to fully address this point, these findings suggest that the behavioral deficit we observed following chemogenetic inactivation is primarily mediated by silencing of excitatory neurons.

We acquired simultaneous LFP recordings from AC and pStr to show that neural activity in these regions is temporally synchronized during the sound cue predicting the interval timing to reward. The activity in AC and pStr was highly synchronized at 0 s lag, which is consistent with the common input received by both these regions from the medial geniculate body (MGB) [80,81,84,85]. In the context of our task, this would suggest that the sound-triggered reward timing encoded in AC and in pStr could be simultaneously and differentially modulated by the neural inputs from MGB. In addition to the direct projections from MGB, pStr receives strong monosynaptic projections from AC [68,69,95]. This is consistent with our finding that in addition to the synchronous activity across these brain regions, a second component of the cross-correlograms points at AC leading the pStr by 38 ms on average. Our LFP recordings in AC and pStr provided a quantification of the overall functional and temporal communication patterns across the AC and pStr. While we focused our analyses on the periods of sound-guided reward time prediction, we cannot determine that the coordination

we observe between the AC and pStr is specific to reward time prediction. Instead, it may be a more general pattern of communication that can facilitate a range of functions. Future studies using single unit recordings could better elucidate how communication across these structures is manifested at the local ensemble level and across different periods and behaviors.

While sound-triggered action timing on the scale of seconds is important for many everyday behaviors, it is a particularly critical ability in humans for verbal communication. In verbal communication, humans use incoming speech sounds to predict when and what sounds are expected to follow [96–99]. Furthermore, the sequential nature of speech sounds helps arrange incoming information in a temporal structure and this ability is impaired when there are hearing deficits [100–104]. Deaf or hard of hearing children face challenges in sequential time perception, which impairs their storytelling ability, a key cognitive development factor [105]. In comparison, children with postlingual cochlear implants exhibit greater improvements in time perception and consequently, their storytelling ability, emphasizing the role of hearing acquisition in sound-guided temporal processing. Other research involving children with mild hearing loss have showcased the benefits of interventions on time sequencing and storytelling [106]. These studies collectively underscore the critical role of sound-guided timekeeping in developing and maintaining language-related skills and highlight the need to further study the neural mechanisms underlying these processes to develop tailored support for individuals with hearing impairments.

## Materials and methods

### Ethics statement

All animal procedures were performed in accordance with the University of Michigan animal care committee's regulations, specifically animal protocols # PRO00009935 and # PRO00011694.

### Animals

We used 60 (40 males, 20 females, 8–30 weeks of age) C57BL/6J mice (Jax number: 000664). Mice were individually housed under a reverse 12 h light/12 h dark cycle, with lights on at 8:30 PM and off at 8:30 AM, and had access to ad libitum food, water, and enrichment. During the behavioral training period, mice were on water restriction and given 2–5 ml of water each day in addition to the water consumed during behavioral training to keep them sufficiently hydrated.

### Surgical procedure

All surgeries were performed on mice anesthetized using isoflurane (1.5%–2% vol/vol). Anesthetized mice were placed in a stereotaxic frame (Kopf 514 Instruments, CA, USA), and an anti-inflammatory drug (Carprofen, 5 mg/kg, subcutaneous injection) and a local anesthetic (lidocaine, subcutaneous injection) were administered. A custom-made lightweight (<1 gr) titanium head bar was attached to the back of the skull using dental cement and cyanoacrylate glue to allow for head-fixed behavior. During the surgery, body temperature was maintained at 38 °C, and the depth of anesthesia was regularly assessed by checking the pinch withdrawal reflex. A small craniotomy was performed over target coordinates relative to the bregma (AC: −2.7 mm anterior from bregma, ±4.3 mm from midline, −0.55 mm ventral, 0° angle; pStr: −1.7 mm anterior from bregma, ±3.35 mm from midline, −2.8 mm ventral, 0° angle).

To chemogenetically target the neural projections from AC to pStr and for pharmacological inactivation experiments using muscimol, custom-made cannulae (25-gauge tubing) or guide cannulas (Plastics One) were placed at the surface of the brain in these craniotomies at the target regions and secured to the skull using dental cement. Dummy cannulae were inserted into these cannulae to prevent outside debris from entering the cannula.

Mice were treated with Carprofen for 48 h post-surgically and were allowed to recover for a week.

## Electrophysiological recordings

Tungsten wire electrodes (two 50 μm wire bundle) were used to acquire local field potential (LFP) responses in AC and pStr. Electrodes were dipped in Di-I dye (Invitrogen, Catalog # 22885) before insertion. The ground screw was positioned over the cerebellum (2.0 mm posterior to lambda, 3.5 mm from midline). Electrodes were unilaterally implanted into AC and pStr on the right hemisphere, (distance between the two electrodes within a region was <50 μm, distance between electrode arrays in AC and pStr was approximately 2.67 mm). LFP signals were acquired using a Tucker-Davis Technologies (TDT) acquisition system and Synapse Lite Software. The output bioelectrical signal was digitized, sampled at 6 kHz, and bandpass filtered in 0.5–300 Hz for LFP recordings. All data acquired was saved for offline data processing.

## Virus injections

Viruses were acquired from Addgene to inactivate the anterograde projections from AC to pStr chemogenetically. To achieve projection specificity in C57BL/6J WT mice, we used an established dual viral approach. We bilaterally injected Cre-dependent DREADD virus (Roth, 2016 [107]): AAV5-hSyn-DIO-hM4D(Gi)-mCherry (2.4E + 13 GC/ml, 350 nl, Addgene catalog # 44362) or AAV5-hSyn-DIO-mCherry (2.6E + 13 GC/ml, 350 nl, Addgene catalog # 50459) into the AC and a retrograde Cre virus: pENN/AAVrg-hSyn-Cre-WPRE-hGH (1.8E + 13 GC/ml, 200 nl, Addgene catalog #105553) into the pStr. The infusions were done using a 32-gauge injection needle (custom length per infusion site) or through thin internal cannulas (Plastics One) inserted into the implanted cannulae in the brain, connected to a 10 μl Hamilton syringe at a rate of 50 nl/min.

## Drug administration

**Muscimol infusions.** Mildly sedated mice were bilaterally infused with 0.5 μg/μl muscimol (BODIPY TMR-X fluorophore-conjugated, ThermoFisher, Catalog Number – M23400) dissolved in phosphate-buffered saline (PBS) and 1.5% DMSO or PBS with 1.5% DMSO as a control (Volume per hemisphere = AC: 750 nl [108,109], pStr: 360 nl [63]) at a rate of 150 nl/min, into cannulae implanted in target sites. The infusions were done via custom-made injectors or thin internal cannulas (Plastics One) as previously described.

**CNO injections.** An amount of 5 mg Clozapine-N-Oxide (CNO) (HelloBio) was diluted in 0.9% saline solution. All animals in the chemogenetic inactivation experiments were first injected with saline (5 mg/kg, i.p.) as a control and then were injected with the prepared CNO (5 mg/kg, i.p.) solution the following day, to chemogenetically inactivate the projections from AC to pStr.

## Behavior

All our behavioral setups were custom built and controlled by an Arduino (Arduino Uno board with an Adafruit Music Maker shield) circuit. Behavioral data acquired through the Arduino IDE software was saved in text files for analysis. Videos of animal behavior were acquired using Logitech C920 HD Pro camera on the LogiCapture software and using the Angetube 1080p web camera on the Bandicam software, under red light conditions.

**Paradigm.** Mice were trained on an appetitive sound-triggered reward time prediction task. In this task, mice on water restriction were head-fixed inside a tube to reduce movement-related artifacts, presented a sound cue from a speaker (4 Ω, 3 W Shutao magnetic speakers with a frequency range of 0–20 kHz, placed approximately 10 cm away from the animal's head on the left) and trained to consume a water reward from a reward port placed close to its mouth. A trial constituted a 1.5 s long sound cue, and a water reward delivery separated by a fixed time interval (0.5–5 s), with randomized inter-trial intervals in the range of 2–6 s (Fig 1A, Trial block). The sound cue was a sequence of three 0.5 s long pure tones (8 kHz, 12 kHz, 16 kHz; 5 ms rise/fall time, presented at 55–60 dB SPL) generated at a 97.5 kHz sampling rate using MATLAB (Mathworks 2019a). We used a sequence of tones as a balance between acoustic simplicity,

the prominence of time-varying sounds in ethological settings, and the known involvement of the AC in encoding such time-varying sounds, including via offset responses [48].

**Training.** Water-restricted mice were handled and habituated to the experimental setup for approximately 7 days. In this period, mice were head fixed and trained to lick the reward port through which a water reward was delivered randomly at 3–10 s intervals without any sound cue. The reward port consisted of a metal tube that delivered a fixed amount of water (approximately 3 µl) each trial, connected to a capacitance-based lick detector that allowed recording lick times. Mice were also familiarized with the sound cue used in behavioral training over the last 3 days of habituation through random sound presentations (approximately 50 times across all 3 days) at 3–10 s intervals without any reward delivery.

Habituated mice started training on trials with a fixed time interval of 1.5 s from sound termination time to reward, with 150–250 trials per daily training session. After 7–10 days of training, catch trials, in which reward was withheld, were randomly introduced 15%–20% of the trials/session. Once the animal learned to predict reward time at 1.5 s interval, it was then trained to use the same sound to predict a different interval of time between sound and reward (sound-reward interval). We trained each animal to predict timed reward using the same sound cue at four sound-reward intervals – 0.5 s, 1.5 s, 2.5 s and 5 s and always trained them to learn the time intervals in this order – 1.5 s → 2.5 s → 5 s → 0.5 s (Fig 1A).

To identify whether mice predicted the sound-reward interval duration from sound onset or sound termination, we tested a subset of mice trained to predict reward at 1.5 s from sound cue, to use a shorter duration sound cue to predict reward. On this testing day, we randomly interspersed 35% of the trials with 1 s long sound cue (a sequence of 8 kHz and 12 kHz pure tones, each 0.5 s long, with 5 ms rise/fall time) to deliver reward at the same time interval – short sound trials (Fig 3A), along with standard 1.5 s long sound cue trials.

**Behavioral training for the pharmacological inactivation experiments.** In these experiments, we used a GABA-A receptor agonist, muscimol, to inactivate AC or pStr to establish their causal role in sound-triggered reward time prediction task. Each of these experiments consisted of two different cohorts of mice. One cohort of animals were trained to predict timed reward at 1.5 s from sound termination (Figs 2, 5, and 6; 1.5 s Delay task) and another cohort of animals were trained on an alternative version of the task where reward immediately followed the sound cue (Figs 2, 5, and 6; No-Delay task).

**Behavioral training for acquiring electrophysiological recordings in AC and pStr.** Mice implanted with wire electrodes underwent the same habituation protocol as described above. In this habituation phase, these mice were habituated to cables plugged to the electrode connectors (Mouser Part # 437-8618700810001101) on their head implants and trained to lick the reward port to consume the water reward. The reward port for these experiments was a tube fitted with an IR sensor to detect licks through beam breaks. We used an IR beam break based lick detector in these experiments to prevent interference of the capacitance-based lick detector with the acquisition of LFP signals. Following habituation, they were trained on the previously described sound-triggered reward time prediction task on the 4 different time intervals between sound and reward in the order 1.5 s → 2.5 s → 5 s → 0.5 s, while their LFP responses in AC and pStr were recorded throughout each daily training session. LFP responses were monitored for movement using video recording and periods of movement were eliminated prior to analysis.

Another cohort of mice implanted with wire electrodes unilaterally in AC underwent similar habituation and were trained to predict reward at 1.5 s sound-reward interval with the regular reward size (approximately 3 µL, S6 Fig, R2) used for all the experiments in this study. Once they showed consistent predictive licking for this reward size, these mice were trained to predict reward at the same sound-reward interval of 1.5 s but with different reward sizes. First with a larger reward size (approximately 6 µL, S6 Fig, R3) and then with a smaller reward size (approximately 1.5 µL, S6 Fig, R1). Predictive licking and LFP responses in AC were recorded throughout each daily training session.

**Behavioral training for chemogenetic inactivation of AC-pStr projections experiments.** We used chemogenetic inactivation of anterograde projections from AC to pStr to identify their role in sound-triggered reward time prediction task. Animals in these experiments started by learning to predict timed reward at 1.5 s from sound (1.5 s Delay task) and then

underwent chemogenetic AC-pStr projection inactivation with CNO injection to test for effect on behavioral performance. Following a 4-day washout period, these animals trained and underwent chemogenetic manipulation on the No-Delay task (Fig 5A, Task timeline).

In a subset of animals in which DREADDs were expressed in the AC-pStr projections, we also simultaneously recorded LFP responses in pStr using tungsten wire electrodes, while they trained on the 1.5 s Delay and No-Delay tasks.

## Data analysis

All analyses were done using custom-written MATLAB (Mathworks 2022a) scripts unless otherwise mentioned.

**Behavioral data analysis.** To quantify the animal's ability to predict reward at a fixed time interval from a sound cue, we extracted lick times per trial and averaged these licks per time bin (time bin = 1 ms for lick data acquired using the capacitance-based lick detector and time bin = 1/20 ms for lick data acquired using the IR beam break- based lick detector) across trials for each daily training session to get a predictive licking response curve. To measure learning, we computed the slope of this predictive licking response curve (MATLAB command: polyfit, degree 1) in the predictive lick period, which was defined differently for rewarded and catch trials, to include the period after the reward delivery time only in catch trials and not in rewarded trials as this would be reward consumption evoked licking, rather than predictive licking for reward. For rewarded trials, the predictive lick period was defined as the time from sound termination to 100 ms prior to reward delivery time for each sound-reward interval. Contrastingly, we used a more conservative definition of the predictive lick period for catch trials, using the period from 200 ms prior to sound termination to the time at which reward was expected for the 0.5 s interval, for all the sound-reward intervals. We used this slope measure in catch trials to ascertain when the animals had learnt to consistently predict reward within each sound-reward time interval. We compared the slopes of the predictive licking curve for catch trials across training days for each sound-reward interval and picked those days for which the slope value crossed the slope threshold of (mean + 3 × standard deviation of slope values across all training days per sound-reward interval). Amongst the training days that satisfied this criterion, the day with maximum slope value was chosen as the "best" behavior day for each sound-reward interval per animal. We decided to focus on these "best" behavior days to compare animals' ability to predict reward timing at different sound-reward intervals as they showed most consistent and robust predictive licking for reward on these days, compared to earlier days of training on each sound-reward interval (S2 Fig). However, we confirmed that our key findings hold true across 4 other data inclusion criteria as well (S3 Fig): (1) considering the last day of training on each sound-reward interval as the "best" behavior day for the interval; (2) considering the penultimate day of training on each sound-reward interval as the "best" behavior day for the interval; (3) using the slope of average predictive licking response curve for the last 30 trials of each session, over the predictive lick period we defined above for the catch trials, to determine the "best" behavior day for each sound-reward interval; (4) using the slope of average predictive licking response curve for a subset of trials (trial # 25–100) of each session, over the predictive lick period we defined above for the catch trials, to determine the "best" behavior day for each sound-reward interval.

We determined the effect of muscimol and chemogenetic inactivation on behavioral performance by comparing the predictive lick responses only in catch trials on control training day (PBS infusion for pharmacological inactivation experiments or saline injections (i.p.) for chemogenetic inactivation experiments) to the predictive lick responses on the manipulation day (muscimol (MUS) infusion for pharmacological inactivation experiments or CNO injections (i.p.) for chemogenetic inactivation experiments). We eliminated animals which did not show a significant change in predictive licking (in the predictive lick period stated below in the formula) compared to baseline period (stated below in the formula) within each behavioral session and then applied the following formula to draw comparisons across animals:

$$\log \text{Predictive licking ratio (PLR)} = \log \frac{\text{Avg. across trials [# of licks in (Predictive lick period - Baseline period)] on PBS or Saline}}{\text{Avg. across trials [# of lick in (Predictive lick period - Baseline period)] on MUS or CNO}}$$

*Where, baseline period = sound onset time – 1900 ms to sound onset time – 1150 ms*
*predictive lick period for delay tasks = reward time – 250 ms to reward time + 500 ms*
*predictive lick period for no-delay task = sound onset time + 750 ms.*

To check whether animals' ability to lick for reward changed based on sound termination time or not, we compared their predictive lick responses in standard and short sound trials using a variation of the PLR described above –

$$\textit{log PLR for Standard vs. Short Sound trials} = \log \frac{\text{Avg. across trials [\# of licks in (Predictive lick period} -\textit{Baseline period})] \textit{ for [STANDARD TRIALS]} +\textit{for [SHORT SOUND TRIALS]}}{\text{Avg. across trials [\# of licks in (Predictive lick period} -\textit{Baseline period})] \textit{ for[STANDARD TRIALS]} -\textit{for [SHORT SOUND TRIALS]}}$$

*Where, baseline period = sound onset time – 1900 ms to sound onset time – 1150 ms, and*
*predictive lick period = sound offset time to reward time − 100 s*

**Electrophysiological data processing and analysis.** Acquired electrophysiological data was extracted using TDTBin2mat script (provided by TDT) and organized to synchronize it with lick response times for each session across sound-reward intervals per animal. Using slopes of the predictive lick response curve of catch trials, the "best" behavior day was determined for each sound-reward interval per animal, as described above. Trials in each training session with no licks in the 200 ms period from sound onset or in the 500 ms period from sound offset (hereafter referred to as no-lick trials) were extracted and further analysis was carried out only on these no-lick trials on the "best" behavior days for each sound-reward interval across animals. Session-wise LFP signals for AC and pStr were filtered for movement-related artifacts by eliminating any signal above a threshold of mean LFP signal for the session + 3 × (standard deviation of the session LFP signal) and then were *z*-scored for each session prior to analysis.

To analyze sound-evoked LFP responses, trial-wise LFP activity in AC and pStr were aligned to the sound onset and baseline-corrected by subtracting its average during the 10 ms period from sound onset for computing the onset response magnitude (similar to [52]) or from sound offset for computing the offset response magnitude. Response magnitude was defined as the amplitude of the maximum trough (most negative) in the 40 ms period from sound onset/offset for each no-lick trial per session. Response magnitudes were averaged across trials per session and normalized to the average response magnitude of the shortest sound-reward time interval (0.5 s) session per animal. Normalized response magnitudes were combined across animals per sound-reward interval and compared using the Kruskal–Wallis test (MATLAB command: kruskalwallis), with a post-hoc Tukey-Kramer test to determine individual group differences (MATLAB command: multcompare applied on the kruskalwallis output).

To compare the pStr onset response magnitudes in saline and CNO conditions, we averaged the response magnitudes across trials and tested for significant differences per animal between the conditions using the Wilcoxon rank-sum test (MATLAB command: ranksum). These average response magnitudes were then normalized to average of the saline condition to determine the population-level trends for the 1.5 Delay and No-Delay tasks.

We examined the coordination in AC and pStr LFP activity during sound by computing the trial-by-trial correlation of sound onset response magnitudes in AC and pStr (MATLAB command: corrcoef). Significant correlation coefficients across animals were combined per sound-reward interval and compared across intervals using the Kruskal–Wallis test (MATLAB command: kruskalwallis), with a post-hoc Tukey-Kramer test to determine individual group differences (MATLAB command: multcompare applied on the kruskalwallis output).

To determine the temporal relationship of AC and pStr LFP responses during sound-triggered reward time prediction behavior, we ran a cross-correlation analysis of the LFP responses in AC and pStr for each session (MATLAB command: xcorr, using the "normalized" model, maxlag = ±100 ms) over a 2 s period from 0.5 s prior to sound onset to

sound termination time. By shuffling the identity of AC and pStr traces per session and computing their cross-correlation coefficients, we simulated chance cross correlation coefficients between AC and pStr sound responses (Number of iterations = 100). We generated difference traces by calculating the difference in average cross correlation coefficients between real and simulated data and then determined the time which these difference traces peaked giving us the time lag between AC and pStr sound responses per training session. We calculated the median of this peak time of difference traces across all animals and sound-reward intervals and compared this median against 0 s using the Wilcoxon signed rank test (MATLAB command: signrank).

**Statistical tests.** We used statistical tests at a $p < 0.05$ significance level and $\alpha = 0.05$ for all comparisons unless otherwise indicated.

**Histology.** Mice were euthanized with an overdose of isoflurane (5%) or carbon dioxide (2%) and perfused transcardially with PBS (0.9%), followed by 10% paraformaldehyde (PFA). The brain tissue was removed and fixed in 10% PFA for 72 h. For cryoprotection, the brain tissue was transferred to 30% sucrose solution for 3–4 days before sectioning. Coronal sections (50 μm thickness) were obtained in a cryostat (Leica) and kept in PBS at 4 °C before mounting. All sections were mounted in glass slides and covered with Fluoroshield mounting medium with DAPI (Abcam, USA). Images of brain sections were acquired using a fluorescent microscope (Zeiss) equipped with an apotome (Olympus) and the ZenPro software.

For all histological validations, brain sections were imaged with a 10× objective, examined for cell nuclei labeling DAPI expression (470 nm), and saved as both multichannel and individual fluorophore channels composite tiff images. Histological validation was done by overlaying brain sections over slice images from the Allen Institute's 10 μm voxel 2017 version from the Allen Mouse Brain Common Coordinate Framework [110]. Histological validation was used as an inclusion criterion for all behavioral, electrophysiological, and chemogenetic experiments.

**Tungsten electrode track verification.** Electrode tracks were identified by DiI expression (550 nm). Images with DiI expression and electrode tracks were saved and further analyzed with the open-source SHARP-Track toolkit [111]. Verified electrode tip coordinates were compared to the coordinate range showing the projections from AC to pStr shown in the Allen Mouse Brain Connectivity Atlas (Allen Mouse Brain Connectivity Atlas, connectivity.brain-map.org/projection/experiment/146858006).

**Reconstruction of virus expression.** The procedure for reconstructing viral expression volumes was similar to electrode track reconstruction. However, the regions defined in each slice are two-dimensional, and the volume is delineated at the 3D projection of all the combined sections showing virus expression. AP, ML, and DV coordinates from all sections with positive mCherry expression were again compared to the range of AC-pStr projections shown in the Allen Mouse Brain Connectivity Atlas (Allen Mouse Brain Connectivity Atlas, connectivity.brain-map.org/projection/experiment/146858006). Animals showing mCherry cell body expression outside AC areas and mCherry axonal expression outside pStr were not included.

**Muscimol infusion.** To verify the muscimol diffusion within the target areas, brain slices were examined for BODYPY-TmX (BDP-T, 573 nm) expression. Brain sections showing positive BDP-T labeling were saved per animal. Further comparison with the reference Allen Atlas and 3D projection showed the AP and ML coordinates with muscimol diffusion. Clear hit for AC or pStr was considered a diffusion only within the target areas. For pStr experiments, animals with muscimol diffusion up to rostral striatal areas (anterior to −1.2 mm from bregma) were not included.

## Supporting information

**S1 Fig. Licking during the baseline period does not change across sound-reward intervals and with manipulations. A.** Boxplot showing the distribution of licking during the baseline period across animals across sound-reward intervals on the best behavior days are not significantly different ($p = 0.251$, Kruskal–Wallis test). In each box, the solid

line indicates the median, and the bottom and top edges of the box indicate the 25th and 75th percentiles, respectively. The whiskers extend to the minimum and maximum values in the dataset. Individual circles represent the average value for each animal and per animal data points across sound-reward intervals are connected by the black dashed lines. **B–E**: Box plot showing the ratio of change in the number of baseline licks from control day (PBS or Saline) to manipulation day (MUS or CNO) across animals trained on the 1.5 s Delay and No-Delay tasks with muscimol infusions in the AC (**B:** $N = 8$, $p = 0.121$, Wilcoxon rank-sum test) and in the pStr (**C:** $N = 5$, $p = 0.841$, Wilcoxon rank-sum test), and with chemogenetic inactivation of the AC-pStr projections in the experimental group (**D:** $N = 8$, $p = 0.295$, Wilcoxon rank-sum test) and in the control group (**E:** $N = 6$, $p = 0.3125$ Wilcoxon rank-sum test). Lines connecting the circles represent the ratio of change in baseline licks for each animal when trained on the 1.5 s Delay and No-Delay tasks. In each box, the solid line indicates the median, and the bottom and top edges of the box indicate the 25th and 75th percentiles, respectively. The whiskers extend to the minimum and maximum values in the dataset.
(TIF)

**S2 Fig. Evolution of sound-triggered predictive licking for each sound-reward interval: Behavioral performance of a representative animal on the first day of training and best behavior day chosen for each of the four sound-reward intervals.** For each session, the four panels of plots show: Top: Peri-sound lick raster of the animal performing on rewarded (left) and catch (right) trials within the session. Bottom: Average peri-sound lick rate (licks/bin, bin = 1 ms) response curve (solid line denotes mean, shaded area represents SEM across trials) for the behavioral session above for rewarded (left) and catch (right) trials. Shaded pink region represents the 1.5 s long sound period. Solid and dotted lines represent when reward was given in rewarded trials and expected in catch trials. Black ticks represent licks. Each color represents the different sound-reward intervals. Note: The plots are shown in the order of training across sound-reward intervals, starting with the 1.5 s delay and ending with the 0.5 s delay. The very first session of training on the 1.5 s delay did not include catch trials.
(PNG)

**S3 Fig. Behavioral and neural encoding of the predicted time to reward holds for different inclusion criteria of days and trials.** The rows represent the four different criteria used to replicate the findings – (Top to bottom): Last of day of training on each sound-reward interval, penultimate day of training on each sound-reward interval, maximum value of the estimated slope for the last 30 trials per session across days of training per sound-reward interval and maximum value of the estimated slope for a subset of trials (25–100) across days of training per sound-reward interval. For each criterion, Column 1–2: show box plots for the estimated slope of predictive licking response curves for each of the four sound-reward intervals across all animals ($N = 8$) for rewarded trials. In each box, the solid line indicates the median, and the bottom and top edges of the box indicate the 25th and 75th percentiles, respectively. The whiskers extend to the minimum and maximum values in the dataset. Individual circles represent the average slope for each animal and per animal data points across sound-reward intervals are connected by the black dashed lines. Column 3–4: have boxplots showing the AC (Col 3) and pStr (Col 4) onset response magnitude computed across animals ($N = 8$) for each of the sound-reward intervals. In each box, the solid line indicates the median, and the bottom and top edges of the box indicate the 25th and 75th percentiles, respectively. The whiskers extend to the minimum and maximum values in the dataset.
(PNG)

**S4 Fig. Histological verification of the bilateral spread of muscimol infusion in AC and pStr A. and C.** Brain slices acquired from an example animal with muscimol infusion sites seen in orange. Markers on either side indicate the depth at which muscimol was infused in the left and right hemispheres. Scale bar: 1000 μm. **B. and D.** Brain slices showing the spread of muscimol in the A/P axis for an example animal for AC and pStr.
(TIF)

**S5 Fig. AC LFP responses to the second and third tones in the sound cue used to predict time to reward changes with training. A–C:** Normalized average AC LFP responses to the three different tone components of the sound cue from an example animal trained on the four different sound-reward intervals represented by the different colors (solid line denotes mean, shaded area represents SEM across no-lick trials). Dashed black lines represent onset of the tone as shown in the illustration above. **D–F:** Boxplot showing the AC LFP response magnitude computed across animals ($N = 8$) for each of the three tones across sound-reward intervals. In each box, the solid line indicates the median, and the bottom and top edges of the box indicate the 25th and 75th percentiles, respectively. The whiskers extend to the minimum and maximum values in the dataset. Comparison across sound-reward intervals yields – First tone: ***$p = 1.42 \times 10^{-36}$, Second tone: ***$p = 1.11 \times 10^{-7}$, Third tone: ***$p = 4.38 \times 10^{-11}$ (Kruskal–Wallis test). **G.** Ratio of the sound onset response magnitudes for (Left box) First to the second tone and (Right box) First to the third tone. (***$p < 0.00001$ compared against 1, Wilcoxon rank-sum test).
(TIF)

**S6 Fig. Predictive licking and AC response magnitudes do not change with varying reward sizes. A.** An Illustration of the behavioral timeline. **B.** Average peri-sound lick rate (licks/bin, bin = 1/20 ms) response curves (solid line denotes mean, shaded area represents SEM across trials) of an example animal trained to perform on the 1.5 s delay task with three different reward sizes represented by different colors. Left: Rewarded trials; Right: Catch trials. Shaded pink region represents the 1.5 s long sound period. Solid and dotted lines represent when reward was given in rewarded trials and expected in catch trials for each of the sound-reward interval. **C.** Box plot representing the estimated slope of predictive licking response curves for each of the three reward sizes across all animals ($N = 4$) for rewarded trials (left, $p = 0.334$, Kruskal–Wallis test) and for catch trials (right, $p = 0.219$, Kruskal–Wallis test). In each box, the solid line indicates the median, and the bottom and top edges of the box indicate the 25th and 75th percentiles, respectively. The whiskers extend to the minimum and maximum values in the dataset. Individual circles represent the average slope for each animal and per animal data points across sound-reward intervals are connected by the black dashed lines. **D.** Normalized average AC LFP responses to the sound onset from an example animal trained on the three different reward sizes represented by the different colors (solid line denotes mean, shaded area represents SEM across no-lick trials). Dashed pink line represents the period the sound onset. **E.** Boxplot showing the onset response magnitudes computed across animals ($N = 4$) for each of the reward sizes. In each box, the solid line indicates the median, and the bottom and top edges of the box indicate the 25th and 75th percentiles, respectively. The whiskers extend to the minimum and maximum values.
(PNG)

**S7 Fig. Histological verification of targeting AC to pStr projections and change in LFP responses in pStr following chemogenetic inactivation of these projections. A.** Histological validation of electrode position in pStr and chemogenetic virus expression in AC to pStr projections. I. Electrode position in pStr is denoted by the green dotted lines. II. and III. Representation of the virus injection tracks in AC and pStr. IV. and V. are magnified images from II and III showing axons in pStr and cell bodies in AC. **B and C.** Normalized average pStr LFP (solid dashed line denotes mean, shaded area represents SEM across no-lick trials) recorded in response to the sound onset from an example animal trained on the 1.5 s Delay task **(B)** and on the No-Delay task **(D)** with saline (orange/light blue) and CNO (black) injections. The shaded pink region represents the period from sound onset. **D and E.** Comparison of the normalized average pStr onset response magnitude computed across animals ($N = 3$) for the 1.5 s Delay task **(C)** and for the No-Delay task **(E)** on saline and CNO conditions. Error bars represent mean ± SEM across animals. Comparison between saline and CNO conditions yields *$p < 0.05$ for each animal when trained on the 1.5 s Delay task and was not significantly different when trained on the No-Delay task.
(TIF)

## Acknowledgments

We would like to thank Lily Allen, Zeinab Mansi, Chloe Cordes, Lindsay Cain, and Grant Griesman for their help with training animals on the task and Siddharth Mohite for his help with setting up the analysis pipeline scripts.

## Author contributions

**Conceptualization:** Harini Suri, Gideon Rothschild.

**Data curation:** Harini Suri, Karla Salgado-Puga, Yixuan Wang, Nayomie Allen, Kaitlynn Lane, Kyra Granroth, Alberto Olivei, Nathanial Nass.

**Formal analysis:** Harini Suri, Karla Salgado-Puga.

**Funding acquisition:** Gideon Rothschild.

**Investigation:** Harini Suri, Gideon Rothschild.

**Project administration:** Gideon Rothschild.

**Supervision:** Gideon Rothschild.

**Writing – original draft:** Harini Suri.

**Writing – review & editing:** Harini Suri, Gideon Rothschild.

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
