## [Editor Report · Decision Letter 0]

Dear Dr Rothschild,

Thank you for submitting your manuscript entitled "A Cortico-Striatal Circuit for Sound-Triggered Prediction of Reward Timing" for consideration as a Research Article by PLOS Biology.

Your revised manuscript has now been evaluated by the PLOS Biology editorial staff as well as by an academic editor with relevant expertise and I am writing to let you know that we are likely to accept this manuscript for publication, provided you satisfactorily address the following data and other policy-related requests. Please note that we could not check for some points at this stage because we need the full submission for this, so there may be further requests down the line.

* We would like to suggest a different title to improve its accessibility for our broad audience: An auditory cortical-striatal circuit supports sound-triggered timing to predict future events

* Please add the links to the funding agencies in the Financial Disclosure statement in the manuscript details.

* DATA POLICY:

Regardless of the method selected, please ensure that you provide the individual numerical values that underlie the summary data displayed in the following figure panels as they are essential for readers to assess your analysis and to reproduce it: 1D, 2D, 3CD, 4CFG, 5GH, 6C, 7CEIJ, S1ABCDE, S3A–L, S5DEFG, S6CE and S7DE.

* CODE POLICY

We expect to receive your revised manuscript within two weeks.

*Published Peer Review History*

*Press*

Sincerely,

Christian

Christian Schnell, PhD

Senior Editor

PLOS Biology

cschnell@plos.org

---

## [Editor Report · Decision Letter 1]

Dear Dr Rothschild,

Thank you for the submission of your revised Research Article "An auditory cortical-striatal circuit supports sound-triggered timing to predict future events" for publication in PLOS Biology. On behalf of my colleagues and the Academic Editor, Mathew Diamond, I am pleased to say that we can in principle accept your manuscript for publication, provided you address any remaining formatting and reporting issues. These will be detailed in an email you should receive within 2-3 business days from our colleagues in the journal operations team; no action is required from you until then. Please note that we will not be able to formally accept your manuscript and schedule it for publication until you have completed any requested changes.

While you attend to those requests to come, please also make sure to include the approval/license number of the ethical approval for the animal experiments in the Methods section of your manuscript.

PRESS

We frequently collaborate with press offices. If your institution or institutions have a press office, please notify them about your upcoming paper at this point, to enable them to help maximize its impact. If the press office is planning to promote your findings, we would be grateful if they could coordinate with biologypress@plos.org. If you have previously opted in to the early version process, we ask that you notify us immediately of any press plans so that we may opt out on your behalf.

Sincerely, 

Christian

Christian Schnell, PhD

Senior Editor

PLOS Biology

cschnell@plos.org